# Transcriptional Silencing of 35S rDNA in *Tragopogon porrifolius* Correlates with Cytosine Methylation in Sequence-Specific Manner

**DOI:** 10.3390/ijms25147540

**Published:** 2024-07-09

**Authors:** Roman Matyášek, Radka Kalfusová, Alena Kuderová, Kateřina Řehůřková, Jana Sochorová, Aleš Kovařík

**Affiliations:** Institute of Biophysics of the Czech Academy of Sciences, 612 65 Brno, Czech Republic; kalfusova@ibp.cz (R.K.); alenakud@ibp.cz (A.K.); rehurkova@ibp.cz (K.Ř.); sochorova@ibp.cz (J.S.); kovarik@ibp.cz (A.K.)

**Keywords:** *Tragopogon porrifolius* ssp. porrifolius, 35S rDNA copy number variations, bidirectional nucleolar dominance, transcriptional silencing/activation, methylation dynamics of CGs, CWGs, CCGs and CHHs

## Abstract

Despite the widely accepted involvement of DNA methylation in the regulation of rDNA transcription, the relative participation of different cytosine methylation pathways is currently described only for a few model plants. Using PacBio, Bisulfite, and RNA sequencing; PCR; Southern hybridizations; and FISH, the epigenetic consequences of rDNA copy number variation were estimated in two *T. porrifolius* lineages, por1 and por2, the latter with more than twice the rDNA copy numbers distributed approximately equally between NORs on chromosomes A and D. The lower rDNA content in por1 correlated with significantly reduced (>90%) sizes of both D-NORs. Moreover, two (L and S) prominent rDNA variants, differing in the repetitive organization of intergenic spacers, were detected in por2, while only the S-rDNA variant was detected in por1. Transcriptional activity of S-rDNA in por1 was associated with secondary constriction of both A-NORs. In contrast, silencing of S-rDNA in por2 was accompanied by condensation of A-NORs, secondary constriction on D-NORs, and L-rDNA transcriptional activity, suggesting (i) bidirectional nucleolar dominance and (ii) association of S-rDNAs with A-NORs and L-rDNAs with D-NORs in *T. porrifolius.* Each S- and L-rDNA array was formed of several sub-variants differentiating both genetically (specific SNPs) and epigenetically (transcriptional efficiency and cytosine methylation). The most significant correlations between rDNA silencing and methylation were detected for symmetric CWG motifs followed by CG motifs. No correlations were detected for external cytosine in CCGs or asymmetric CHHs, where methylation was rather position-dependent, particularly for AT-rich variants. We conclude that variations in rDNA copy numbers in plant diploids can be accompanied by prompt epigenetic responses to maintain an appropriate number of active rDNAs. The methylation dynamics of CWGs are likely to be the most responsible for regulating silent and active rDNA states.

## 1. Introduction

35S rRNA genes (rDNA) are one of the most abundant repetitive elements in eukaryotic genomes. Hundreds to thousands of rDNA copies are typically organized into tandem arrays, the so-called nucleolar organizing regions (NORs) [1,2]. Its repetitive nature makes rDNA one of the most unstable genomic regions, where unequal recombination may result in intragenomic variability in rDNA copy numbers [3,4,5,6,7], which may play an important role in maintaining genomic integrity, gene expression diversity, and evolutionary adaptability [8,9,10]. Furthermore, recombination plays an important role in sequence homogenization when individual rDNA units in an array evolve concertedly, resulting in the co-existence of numerous nearly identical units in the same genome [11]. Regardless, multiple rDNA variants can be simultaneously present in the same genome and even within a single rDNA array [12,13,14,15], and different rRNA variants can be integrated into ribosomes in a tissue-specific manner [16].

The rDNA is transcribed by RNA polymerase I (Pol I) into a single pre-rRNA transcript formed of 18S, 5.8S, and 26S rRNAs; two internal transcribed spacers (ITS1 and ITS2); and two external (3′ETS1 and 5′ETS2) transcribed spacers, which is then processed into the 18S, 5.8S, and 26S rRNA mature forms [17]. Each transcribed region in a tandem array is separated from both neighboring ones by long, highly variable “non-transcribed” spacers (NTS), which form, together with ETS1 and ETS2, the so-called intergenic spacer (IGS). The occurrence of conserved structural features such as PolI promoter(s), the transcription initiation site (TIS), the transcription termination site (TTS), and tandem repeats indicate that the IGS is a functionally important region participating in the control of Pol I transcription [18]. All these common structural elements within IGS may be arranged in highly complex patterns, resulting in variations between related species, populations, and even within an individual [13,14,15]. 

Because the cellular demand for protein synthesis varies during development, only subsets of rRNA genes are active, and their transcription is under epigenetic dosage control [19]. In genetic hybrids, one manifestation of dosage control is nucleolar dominance (ND), an epigenetic phenomenon in which the rDNAs of one progenitor are repressed [20]. Active rDNAs exhibit open euchromatic structures, while silent rDNAs show a more compact heterochromatic structure, and an intermediate chromatin configuration indicates the rDNAs primed for transcriptional activation [21]. The interplay of DNA methylation, histone modification, and chromatin remodeling directed by non-coding RNAs, often derived from IGS, may be required to establish specific transcriptional states at individual rDNA units [22,23,24,25,26,27]. The ratio of active to inactive rDNAs is often tissue-specific and is stably propagated through cell cycle progression [16,28]. Although the mechanism and functional relevance of rDNA silencing are well understood, little is known about how and which genes are chosen for inactivation [29]. 

DNA methylation at the 5′-cytosine, mediated by a spectrum of DNA methyltransferases, is the most widespread epigenetic mark of DNA in eukaryotes with diverse functions, such as heterochromatin formation [30], transposon silencing [31], mammalian X-chromosome inactivation [32], and gene imprinting [33]. In plants, cytosine methylation may occur in all sequence contexts, CG, CHG, and CHH (H = A, C, or T) [31]. However, the mechanisms responsible for targeting 5 mC in specific sequence contexts and the mutual roles of the individual methylated motifs in rDNA silencing/activation are not fully understood. The methylation pattern can be guided by specific histone modifications [34,35], the pairing of repetitive sequences [36], small RNA interactions [37], and transcription factors [38]. Active DNA demethylation, mediated by DNA glycosylases is crucial for genome-wide epigenetic reprogramming and locus-specific gene activation during plant development [39]. 

*Tragopogon porrifolius* (2n = 2x = 12) is a polyphyletic, morphologically highly variable diploid species composed of at least three subspecies, native throughout the Mediterranean [40]. *T. porrifolius* ssp. porrifolius, phylogenetically well separated from other collections, has a karyotype possessing two NORs on chromosome pairs A and D, while the remaining subspecies, australis and cupani, have only one NOR on chromosomes A, as is typical of most species of the genus [41,42,43]. 

To characterize the genetic and epigenetic relationships between A- and D-NORs in *T. porrifolius* ssp. porrifolius (hereafter referred to as *T. porrifolius*), we used two highly related por1 and por2 lineages (Appendix A) that substantially differed in rDNA copy numbers and relative sizes of A- and D-NORs. Based on these differences, global repetitive organization of IGS (PacBio sequencing), transcriptional activity (RNA-seq, cDNA-CAPS), higher-order chromatin structure (Fluorescence In situ Hybridizations), and cytosine methylation (Bisulfite-Illumina sequencing) were assigned to both A- and D-NORs. Furthermore, the genetic and epigenetic consequences associated with variations in rDNA copy numbers were evaluated. Finally, methylation dynamics of individual sequence motifs in multiple genetically highly related rDNA variants, both transcriptionally active and silent, were used for the estimation of the relative participation of different cytosine methylation pathways in the regulation of rDNA transcription.

## 2. Results

### 2.1. One Abundant rDNA Variant Was Detected in por1, Whereas Two rDNA Variants Occurred in por2

We used long PacBio sequences to analyze the structure variability of rDNA units in the por1 and por2 lineages of *T. porrifolius*. The IGSs in both lineages were built of common sequence motifs mainly represented by multiple kinds of tandemly arranged repeats (R) and dispersed promoter elements (Ps) (Figure 1a). The most abundant repeat R_103_ was composed of 103–104 bp long, equally oriented repetitive units distributed across multiple (3–5) blocks of variable sizes (1–12 R_103_ units). Each block of R_103_ repeats was flanked by Ps (~80 bp) that shared the motif TATATATRGGG closely related to plant TIS for PolI. The region covering P_5_ and R_103c_ was frequently deleted, increasing the structural variability of IGS (Appendix A). Sequence similarities between individual Ps were considerably high (>95%) (Appendix A). Arrays of R_103_ and P repeats were mostly interrupted by two microsatellites (R_2_, R_7_), and one satellite (R_156_). R_2_ and R_7_ were represented by partially degenerated (AT)_0–50_ and (C_2_T_2_G_3_)_7_ motifs, respectively, the latter with the propensity to form G4 structures. The satellite R_156_ was mostly formed from four 152–158 long units. These three repeats were seldom deleted, resulting in rarely occurring truncated IGSs (Appendix A). Degenerated dimers of both 98 bp and 16 bp long regions and a short CpG island were detected in ETS2. 

Alignment of PacBio reads (Figure 1a), PCR amplifications (Figure 1b), and Southern hybridizations (Figure 1c) consistently revealed two major rDNA variants (called L and S) in por2, while only S-rDNA variant was found in the por1. The most noticeable differences between both variants were (i) the relative lengths of the R_103b_ and R_103d_ repeat arrays (Figure 1a) and (ii) two abundant T>G substitutions in ETS2 (Figure 1e). Despite different lengths, the R_103bL_ and R_103bS_ satellites were highly related, differing mainly in the insertion of the R_103_ dimmer formed of h- and g-R_103_ variants. In contrast, repeat arrays R_103dL_ and R_103dS_ differed in all repeat units. Semiquantitative evaluations of hybridization signals (Figure 1c) and sequencing (Appendix A and Appendix A) revealed that (i) L-rDNA in por2 was substantially more abundant than S-rDNA in both por1 and por2 lineages and (ii) por2 lineage contained more than twice as much rDNA as por1.

On the phylogenetic tree, the S- and L-rDNA variants formed two well-supported clades (Figure 1d). Each clade comprised several subclades, indicating the presence of at least five abundant subvariants in S-rDNA (S_1_–S_5_) and two subvariants in L-rDNA (L^a^ and L^s^). One prominent G>A transition found within the target for NlaIII (Figure 1a,e) distinguished L^a^- from both S- and L^s^-rDNAs. Four abundant nucleotide substitutions distinguished S-rDNA subvariants, whose mutual proportions substantially differed between por1 and por2 (Figure 1f and Appendix A). 

### 2.2. Multiple rDNA Variants within Each L- and S-rDNA Array Differed in Transcriptional Activity

To determine the relative transcription activity of individual rDNA variants, we carried out transcriptome analyses by RT-CAPS and RNAseq. To distinguish L- and S-rDNA variants by CAPS, we used MboI having two targets in the L-rDNA and a single target in the S-rDNA (Figure 2a). In por2, genomic CAPS visualized two bands corresponding to L- and S-rDNAs, while there was a single fragment derived from the S-rDNA in the por1 lineage. At the transcriptome level, the S-rDNA variant was amplified in por1, whereas only the L-rDNA variant was amplified in por2. There was no amplification of the S-rDNA variant in this lineage. Consistent with the RT-CAPS analysis, only L-rDNA specific transcripts (both MboI targets) were detected in the por2 transcriptome by RNAseq (right bar charts). To distinguish L^a^-rDNAs from both L^s^- and S-rDNA variants by CAPS, we used NlaIII having one target in both L^s^- and S-rDNAs but no target in L^a^-rDNA (Figure 2b). In por2, genomic CAPS visualized undigested PCR products corresponding to L^a^-rDNA and two digested fragments corresponding to a mixture of S- and L^s^-rDNAs, while there were only digested fragments derived from the S-rDNA in the por1 lineage. At the RNA level, the S-rDNA variant was amplified in por1, whereas only the L^a^-rDNA variant was amplified in por2. There was no amplification of the L^s^ and S-variants in this lineage. Consistent with the RT-CAPS analysis, only specific transcripts from L^a^-rDNA (no target for NlaIII) were detected in the por2 transcriptome by RNAseq (right bar charts). Considering the entire rDNA unit, the number of other polymorphic sites that distinguished L- and S-rDNA variants in the por2 confirmed the selective silencing of the S-rDNAs (Figure 2c). All rDNA sequence motifs were detected in the transcripts but with extremely variable frequency. The most abundant transcripts originated from 18S and 26S rRNA genes, followed by substantially lower contents of transcripts from ETS2 and ITS regions. An even lower transcript content was associated with ETS1 and all seven core promoters. All transcripts were more abundant in root tips than in leaves. 

In contrast to the por2 lineage, the S-rDNAs were transcriptionally active in por1 (Figure 2a,b). The three most abundant S-rDNA variants (S_1_–S_3_) (Figure 1f) were also frequently detected in the transcriptome, although in different proportions than in the genome (Figure 2d,e). Significant enrichment in the transcriptome was associated with under-represented S_2_-rDNA, indicating higher specific transcription than S_1_-rDNA. An elevated proportion in the transcriptome, although statistically insignificant, was also detected for S_3_-rDNA, the third most abundant variant in por1. Both the least abundant S_4_- and S_5_-rDNA variants were transcriptionally silenced as follows from their relative contents in genomes and transcriptomes. 

To estimate tissue-specific variations in the transcriptional efficiency of 35S rDNA more precisely, the contents of ETS2 and 18S rRNA transcripts were related to the total RNA in young leaves, root tips, and five-day-old seedlings (Figure 2f). Compared to leaves, the RNA transcripts in root tips were significantly (*p2* < 0.01) enriched by 18S rRNA if normalized to the total RNA. Enrichment was also detected for seedlings, but only with moderate statistical support (0.05 > *p2* > 0.01). Only root tips showed a moderate enrichment of ETS2 primary transcripts relative to both young leaves and seedlings if normalized to total RNA or 18S rRNA. Because S- and L^s^-rDNA variants were silenced in all examined por2 tissues (Figure 2a–c and Appendix A), we supposed that tissue-specific transcriptional changes were associated with (epigenetic) changes within L^a^-rDNA.

### 2.3. Estimation of the Positions of Individual rDNA Variants within Chromosomal Complement

The distribution of rDNA on individual chromosomes and chromatin condensation was analyzed in root tips by fluorescence in situ hybridization (FISH) using 18S rDNA and 5S rDNA probes (Figure 3 and Appendix A). In the por1 metaphases, the 18S rDNA probe hybridized to two large sites on a submetacetric A-chromosome pair and two faint sites on a pair of D-chromosomes (Figure 3b and Appendix A). On both A- and D-chromosomes the signals were clustered at subtelomeric positions. The 5S rDNA probe hybridized to four interstitial sites, two of which were localized on A-chromosomes. In the por2 metaphases, the 18S probe hybridized to four strong sites on chromosomes A and D (Figure 3d and Appendix A). The signals on the D-chromosomes were stronger than the signals on the A-chromosomes. The 5S rDNA probe hybridization patterns were similar to those in por1.

The secondary constrictions, a hallmark of transcription activity of 35S rDNA, were observed during prophase, on both A-chromosomes in por1 (Figure 3a and Appendix A), while they were located on both D-chromosomes in por2 (Figure 3c and Appendix A). In the por1 interphase, most 18S rDNA signals were located around the nucleolus exhibiting variable levels of chromatin condensation (Figure 3e). In por2, only a fraction of 18S rDNA was located around the nucleolus while two highly condensed signals occurred outside the nucleolus (Figure 3f). Thus, these results collectively indicated that bidirectional nucleolar dominance, associated either with transcriptionally active A-NORs in por1 or with dominant D-NORs in por2, occurred in *T. porrifolius*. The relative sizes of individual rDNA loci (Figure 3g, h) and their epigenetic (condensed/decondensed) patterns were similar in all analyzed individuals within each por1 and por2 lineage. 

Owing to highly related sequences of S-rDNA and L-rDNA variants, it was not possible to assign individual rDNA variants to corresponding NORs by FISH, using specific probes. However, indirect assignment of S-rDNA to A-NORs and L-rDNA to D-NORs was performed based on correlations between (i) the total number of in-size similar NORs (Figure 3 and Appendix A) and the total number of abundant rDNA variants (Figure 1c) in por1 and por2 lineages, (ii) mutual magnitudes of A- and D-NORs and proportions of S- and L-rDNAs in por2 (Figure 3h), (iii) chromatin condensations of both A-NORs in por2 (Figure 3c and Appendix A) and the silent state of S-rDNA in this lineage (Figure 2e), (iv) secondary constrictions on both D-NORs and transcriptional dominance of L-rDNA in por2 (Figure 3a,c,d and Appendix A). As it is widely accepted that chromatin structure and transcription of rDNA are controlled by specific epigenetic markers, we further analyzed differences between individual rDNA variants in selected epigenetic modifications. 

### 2.4. Transcriptional Variability between Individual rDNA Variants Correlated with Their Cytosine Methylation, Particularly in CWG Motifs

To assess the role of individual epigenetic modifications in silencing S-rDNA transcription, por2 seedlings were treated with 5-azacytidine (AzaC), 9-(2,3-dihydroxypropyl) adenine (DHPA), and sodium butyrate (NaBT) known to selectively interfere with respective biochemical pathways. The relative transcription of individual rDNA variants was estimated using cDNA-CAPS (Appendix A). S-rDNA variants were partially activated during germination by DNA-hypomethylating AzaC and DHPA, although AzaC was substantially more efficient than the latter. NaBT, an inhibitor of histone deacetylase [44], had negligible or no effect on activation, although associated morphological changes, represented predominantly by delayed growth of the primary root, were similar to those induced by AzaC (Appendix A). Co-treatment with AzaC and NaBT did not lead to additive or synergistic effects. Because it was difficult to distinguish between drug-activated and persistently active S-rDNAs, the por1 lineage was not subjected to analogous treatments.

To comprehensively characterize the participation of individual methylated cytosines in the regulation of rDNA transcription in *T. porrifolius*, numerous rDNA variants, differing in specific transcriptional activity, were subjected to bisulfite conversion in combination with Illumina sequencing. We focused on the 5′-ETS2 region representing the 5′- end of the primary transcript (Figure 1a) and containing homogenized specific SNPs allowing discrimination between individual rDNA variants (Figure 4a and Appendix A). Two G>T transversions discriminated between major L- and S-rDNA variants. The G>A transition differentiated between L^a^- and (S + L^s^)-rDNA variants on the plus (G-rich) strand. In addition, several other SNPs discriminated between individual S-rDNA variants in por1 (Figure 1f and Figure 2e). To estimate the statistical support of differences in methylation patterns between individual rDNA variants we used: (i) eight biological samples that differed substantially in overall rDNA transcription efficiency (Appendix A, sheet 1), (ii) methylation analyses of each respective cytosine (Figure 4) and (iii) methylation analyses of whole Illumina read (Appendix A). 

An initial comparison of all active and all silenced rDNA variants showed statistically well-supported enrichment of methylated CG or CWG motifs, but not CCGs and CHHs, in silenced rDNA variants compared to active ones (Appendix A, Sheet 2B). These differences were more pronounced in seedlings and root tips than in leaves. Despite the higher average methylation of CGs (97.3%) than CWGs (88.0%), the absolute differences between the methylation of silent and active rDNAs in root tips and seedlings were substantially greater for CWGs (D_s-a_ > 10.5%) than for CGs (D_s-a_ < 1.7%), suggesting that the methylation dynamics of CWGs are likely to be most responsible for regulating active and silent rDNA states in *T. porrifolius*.

To support this assumption, methylation patterns were compared between several pairs of rDNA variants with well-estimated relative transcriptional efficiency: (i) Silent S-rDNAs from por2 were compared with active L-rDNAs from por2 or active S-rDNAs from por1 (Figure 4b and Appendix A). (ii) Silent (S + L^s^)-rDNAs from por2 were compared with active L^a^-rDNAs from por2 or active S-rDNAs from por1 (Appendix A). (iii) Silent L^s^-rDNAs were compared with active L^a^-rDNAs (Appendix A). (iv) On average, less-active L-rDNAs from por2 were compared with, on average, more-active S-rDNAs from por1 (Figure 4b and Appendix A). (v) Active L^a^-rDNAs from por2 were compared with active S-rDNAs from por1 (Appendix A). (vi) On average, less-active S_1,4,5_-rDNAs were compared with, on average, more-active S_2,3_-rDNAs in por1 (Appendix A). (vii) Silent S_5_-rDNAs were compared with active S_2_-rDNAs or S_1,3,4_-rDNAs in por1 (Appendix A). (viii) Less-active S_1,3,4_-rDNAs were compared with more-active S_2_-rDNAs in por1 (Appendix A). 

All pairwise comparisons revealed that each CWG, but not each CG, CCG, or CHH, was more densely methylated in rDNA with a lower transcription efficiency (Appendix A). It is necessary to emphasize that the substantially lower methylation of all CWGs in S-rDNAs from por1 than in L-rDNAs from por2 (Appendix A) also correlated with higher specific activity of S-rDNA in por1 than L-rDNA in por2. Considering the comparable demands for the number of active rRNA genes in the highly related por1 and por2 lineages and the substantially lower content of S-rDNA in por1 than L-rDNA in por2 (Appendix A), the specific activity of S-rDNA in por1 was higher than the specific activity of L-rDNA in por2. All these differences were always most pronounced in seedlings but negligible in leaves. The vast majority of CGs were also methylated more densely in transcriptionally less-active rDNA variants; however, the absolute differences were substantially lower (up to 10%; Appendix A) than in the case of CWGs (up to 46%; Appendix A). In contrast, CHHs and the outer cytosine in CCGs showed often higher methylation in the transcriptionally more-active rDNA(s), with the most illustrative example represented by the S_2,3_- and the S_1,4,5_-rDNA variants in por1 (Appendix A). 

These results, based on the comparison of methylation of individual Cs, were further supported by statistical evaluations of methylation differences between individual rDNA variants within a given tissue as well as between individual tissues for a given rDNA variant (Figure 4 and Appendix A). Similar methylation behavior of both symmetrical CG and CWG motifs were characterized by statistically significantly (*p2* < 0.01) supported enrichment of methylation (Figure 4) in the following: (i) S-rDNAs compared to L-rDNAs, both from por2 root tips and seedlings, (ii) L-rDNAs from por2 compared to S-rDNAs from por1 root tips and seedlings, (iii) S-rDNAs from por2 compared to S-rDNAs from por1 root tips and seedlings, and (iv) leaves compared to root tips (seedlings) for transcriptionally active rDNAs (S from por1 or L from por2) but not for silent rDNA variants (S from por2). Substantially larger absolute differences between average methylations were detected for CWGs than for CGs in all pairwise comparisons. In contrast, no pairwise comparison detected statistically significantly supported differences for CAAs or CCGs, either between individual rDNAs or between individual tissues. However, AT-rich CHHs (CAA, CTA, and CAT), as well as CCGs, showed an even wider overall range of methylation than the CWGs (Appendix A, sheet 2B). Consistently, more than 70% of the pairwise differences between the methylation ranges of individual CAAs, CTAs, or CCGs were supported statistically significantly (Appendix A). However, these proportions were substantially reduced (<30%) for the CGs and CWGs. These analyses indicate that the methylation of individual AT-rich CHHs and perhaps also CCGs was influenced by their spatial position within ETS2. In contrast, the overall methylation variability of CWGs and, to a lesser extent, CGs is mainly due to global epigenetic differences between individual rDNA variants, mainly between active and silent, and also between active rDNAs in different tissues. 

## 3. Discussion

### 3.1. Bidirectional Nuclear Dominance in Non-Hybrid Diploid Tragopogon porrifolius *ssp.* Porrifolius

The chromosomal complement (2n = 2x = 12), distribution of NORs on chromosomes A and D (Figure 3 and Appendix A), flower morphology (Appendix A), and the highly related ETS2 indicated that por1 and por2 lineages are considerably more closely related to each other compared to other *T. porrifolius* accessions. The closest relatives are the accessions from *T. porrifolius* ssp. porrifolius [45,46]. The approximately 150 diploid *Tragopogon* species can be classified into 3 karyotypes differing in the localization of NORs on (i) A- and D-chromosomes, (ii) only A-chromosomes, and (iii) only D-chromosomes [42]. Highly homogeneous ETS2 sequences did not provide evidence that the por1 or por2 arose from hybridization between progenitors with the last two karyotypes. Of course, the original differences between A- and D-NORs, if inherited from possible interspecific hybridization, may have been obscured by homogenization during the evolution of *T. porrifolius*. More likely, the elevated number of NORs in *T. porrifolius* and several other diploid *Tragopogon* species resulted from translocation events [41,46]. Thus, *T. porrifolius* represents a rare example of a non-hybrid diploid with a well-established ND, as indicated by the secondary constriction on D-NORs accompanied by condensation of A-NORs and, at the molecular level, selective silencing of one of two abundant rDNA variants distinct in IGS structure. The dominant character of D-NORs can be preserved even during their silenced state in allotetraploid *T. mirus*, where A^d^-NORs inherited from the *T. dubius* progenitor dominate both A^p^- and D^p^-NORs inherited from the *T. porrifolius* progenitor [47]. Deleting large portions of both A^d^-NORs induced selective activation of only D^p^-NORs [48]. The selective silencing of S-rDNA characterized by a shorter satellite located upstream of the TIS in *T. porrifolius* is most reminiscent of diploid *Hordeum vulgare*, where the activity of the two NORs positively correlates with the number of the 135 bp repeats located upstream of the promoters [49,50].

If both negligible D-NORs in por1 arose as a consequence of the homologous deletion of a large portion of the L-rDNAs, then it is likely that this deletion induced such epigenetic changes in the transcriptionally activated A-NORs that lead to S-rDNA epigenetic variants with a methylation pattern similar to that in dominant L^a^-rDNA in por2. The most pronounced hypomethylation occurred in CWGs and to a lesser extent in CGs, predominantly in seedlings and root tips. At the same time, no or negligible changes in methylations were associated with other motifs. These changes suggest (i) the functionality of some silenced S-rDNA in por2, (ii) similar mechanisms leading to epigenetic patterning in active NORs in por1 and por2, and (iii) the occurrence of bidirectional ND in *T. porrifolius*. Of course, D-NORs in por2 could have arisen also through extensive amplification of D-NORs in por1. However, this direction of rDNA evolution seems less likely given that DNA breaks are more likely in long rDNA arrays [48], and amplifications to such an extent have not yet been observed. The following mechanisms that may be involved in the selective silencing/activation of different rDNA variants (NORs) in *T. porrifolius* can be envisioned.

#### 3.1.1. Competition between Promoters (Enhancers) for Limited Regulatory Elements

The non-hybrid feature of the diploid *T. porrifolius* genome suggests a relatively homogeneous population of transcription factors (TF) involved in 35S rDNA transcription. However, even very small sequence variations between rDNA genes can affect nucleosome positioning or base pairing with noncoding RNAs, thus inhibiting or facilitating TF action [51,52]. Because no structural differences were found between the respective core promoters in both abundant rDNA variants in *T. porrifolius,* we assume that the satellites that are located between the P_2_ and P_3_ promoters and represent the most significant differences between L- and S-rDNAs may bind TF with an efficiency proportional to its lengths, as proposed for *Xenopus* [53,54,55,56] and *Cucumis* [57]. In por2, a limited pool of certain TF can be exhausted by the longer repeat in the more abundant L-rDNA, which are dispersed within the nucleolus available for transcription, while S-genes with a shorter repeat and lower affinity to TF are silenced (Figure 5). In contrast, the tiny D-NORs in por1 left enough transcription factors to activate the A-NORs. In plants, correlations between transcriptional efficiency and the length of tandem repeats surrounding promoters were detected in diploids *Hordeum vulgare* [50]; *Vigna radiata* [18]; *Solanum* [58]; and the allopolyploids *Triticum-Aegilops* [59], *Triticum-Secale* [60], and *Tragopogon mirus* [48,61]. However, no such correlations were detected in artificial *Solanum* allopolyploids [62], *Brassica* [63], and *Arabidopsis* [64], suggesting that other mechanisms may participate in selecting rDNA variants for silencing.

#### 3.1.2. Intact D-NORs Mediate Silencing of A-NORs

The correlation between the silencing of A-NORs and the integrity of D-NORs in *T. porrifolius* suggests that intact D-NORs can trigger the transcriptional silencing of A-NORs. It is likely that some *in trans* acting factors, generated by intact transcriptionally active NORs in por2, can selectively compete with the transcriptional machinery of A-NORs, leading to silencing and increased cytosine methylation, particularly in CWGs (Figure 5). A drastic reduction in the copy numbers of rDNA in D-NORs can completely interrupt the production of these factors, leading to activating A-NORs. A similar scenario can be envisaged for *T. mirus* [47,48] (see above). The activation of silent 35S rRNA genes was recently described in *A. thaliana* mutants with artificially reduced copy numbers of rDNA [65]. Activating fully functional rDNAs, currently, facultative heterochromatin appears to represent a common mechanism for eliminating the fatal consequences triggered by various biotic or abiotic stimuli. The more stable heterochromatic locus serves as a reserve that can be mobilized relatively quickly after disruption of the epigenetic interactions between the loci. 

Although the nature of the “silencers” is largely unknown, specific non-coding RNAs associated with D-NORs may be the most likely candidates. Highly variable rDNA intergenic transcripts are likely involved in rDNA silencing in the mouse [26] and selective silencing of *A. thaliana*-derived rDNA homeologs in the allotetraploid *A. suecica* [25,66]. In *A. thaliana*, specific IGS transcripts from minor rDNA variant(s) might generate siRNA for RNA-directed de novo methylation, which could be involved in silencing the major rDNA variant [67,68]. In this context, it is necessary to emphasize that IGS transcripts in por2 were derived exclusively from D-NORs, although steady-state levels were extremely low. Furthermore, unrelated sequences inserted near the PolI promoter may produce artificial ncRNA that can act as specific silencers, similar to a recently proposed mechanism in animals [69]. 

#### 3.1.3. Mutual Locations of Particular rDNA Variants within NORs

In *T. porrifolius*, several rDNA variants that differed genetically and epigenetically were assigned to each A- and D-NOR. Due to the considerably long and repetitive nature of the rDNA unit, their actual organization within each NOR is currently unknown. This can only be speculated by analogy with several already properly analyzed plant models. In *Oryza sativa,* the rDNA units arranged at the distal end of a single NOR are formed of more copies of the sub-repeats in the IGS than units located proximally [70]. Each of the three prominent rDNA variants in *A. thaliana*, differing in length of 3′-ETS, is clustered over distances as large as 1.5 Mbp along the NOR4. The cluster with the longest 3′-ETS is located at the distal end [12]. In *Allium cernum*, rDNAs with the same number of the A-subrepeats within IGS are clustered into blocks [71]. 

Thus, we suppose that in *T. porrifolius*, individual genetically and epigenetically highly related rDNA variants are clustered rather than intermingled within each NOR (Figure 5). Such arrangement evolved via concerted evolution, resulting in homogenization and mutation spreading among neighboring rDNA units, a mechanism recently proposed by Wang et al. [72]. This process can be accompanied by gradual epigenetic modifications [73,74]. Transcriptionally active and hypomethylated, particularly at CWG motifs, epigenetic variants probably occupy a central decondensed region(s), whereas transcriptionally silent and highly methylated variants form highly condensed boundary knobs. This is consistent with the ultralong sequencing of rDNA in *A. thaliana*, which revealed that most rDNA activity occurs in the central region of NOR4 [75]. These observations suppose that selective rDNA silencing is regulated on a multimegabase scale to inactivate large clusters of rDNA within each NOR rather than by mechanisms dependent on subtle gene sequence variation. These three models are not necessarily exclusive; the action of silencers, a lack of TF, and the positional effect can simultaneously lead to the induction of specific epigenetic states across each NOR. 

### 3.2. Molecular Mechanisms Leading to Sequence-Dependent Relationships between Cytosine Methylation and rDNA Transcription

Correlations between transcriptional silencing and overall cytosine methylation in rDNA variants from *T. porrifolius* are consistent with the generally accepted view that elevated cytosine methylation is associated with rDNA silencing [76]. However, correlations between the methylation of a given cytosine and the transcriptional activity of corresponding rDNA variants in *T. porrifolius* are substantially influenced by the surrounding sequence contexts. The most pronounced correlations were detected for CWGs, consistent with methylation dynamics in the moss *Physcomitrium patens*, where CHG methylation appeared to be a stronger transcriptional suppressor than CG methylation [77], even though CWGs and CCGs were evaluated together. We assume that the variable methylation dynamics of individual C-motifs in *T. porrifolius* can be at least partially controlled by the fine-tuning of individual methylation pathways, which are known to be more or less specific to individual sequence contexts (Figure 5) [35,78]. The average methylation of CGs (97%), CWGs (88%), CCGs (35%), and CHHs (7%) in ETS2 from *T. porrifolius* matched the order found in other plants [79], suggesting that most DNA methylation pathways common in seed plants were conserved in *T. porrifolius*. 

DNA methylation is dynamically controlled by de novo methylation, maintenance methylation, and active demethylation [80]. The de novo DNA methylation in all sequence contexts is mediated by DOMAINS REARRANGED METHYLTRANSFERASE 1/2 (DRM1/2), which is guided to targets via an RNA-directed DNA methylation (RdDM) pathway [37,81]. The increased affinity of DRM2 for CAs and CTs compared to CGs [82] may be related in some way to the more pronounced differences in CWG methylation compared to CG methylation detected between silent and active rDNA variants in *T. porrifolius*.

The methylation of CGs is maintained by METHYLTRANSFERASE 1 (MET1) [83], which methylates hemimethylated CpG sites during replication so that methylation of CGs is restored during the S-phase [84]. MET1 can also interact with HISTONE DEACETYLASE 6 (HDA6), which is required for MET1-dependent CG methylation in a locus-specific manner and for silencing of rDNA [23,85]. These participants in CG methylation may be regulated in a manner leading to statistically supported correlations between cytosine methylation and rDNA silencing. However, because Met1 activity is rather independent of the heterochromatic repressive mark H3K9me2 [35], absolute differences in CG methylation between active and silent rDNA can be kept low, as observed in *T. porrifolius*.

In contrast, methylation in CHGs is maintained through a positive feedback loop in which SUPPRESSOR OF VARIEGATION 3–9 HOMOLOGs 4, 5, and 6 (SUVH4/5/6) recognize CHG methylation and dimethylate histone H3, resulting in heterochromatic mark H3K9me2, which is subsequently recognized by CHROMOMETHYLASE 3 (CMT3). This methylates unmethylated CHGs during the late G2 phase [86]. CHG methylations can, therefore, be kept out of rDNA by a histone demethylase, IBM1, which removes methylation from H3K9me2 [87]. Thus, heterochromatic/euchromatic proportion in rDNA from *T. porrifolius* can be regulated by finely orchestrated regulation of SUVH methylases and IBM1 demethylase followed by promoting CMT3, which may lead to the strongest correlation between the methylation of the CWG motifs and rDNA silencing. Another striking feature of the CWG methylation dynamics described here is substantially more pronounced hypomethylation in seedlings and root tips than leaves, albeit only in transcriptionally active rDNAs. This is consistent with *A. thaliana,* where global CG methylations were 1% higher in shoots relative to root tips, whereas CHG methylations were 5% higher in shoots relative to root tips [88]. This can be partially explained by a model that assumes that CG (and partially also CCG) methylations are rapidly restored during the S phase by MET1, whereas CWG methylations require H3K9me2 deposition and, therefore, must occur in late G2 [84], resulting in a sharp reduction in methylated CHGs in rapidly dividing cells such as root tips and seedlings, but not young leaves. 

The methylation dynamics of the CCGs characterized by (i) substantially lower overall methylation, consistent with whole-genome methylation analyses in *A. thaliana* [89]; (ii) no correlation between methylation and silencing; and (iii) a higher positional effect indicate that CWG and CCG methylations in rDNA from *T. porrifolius* are regulated by different mechanisms (a) CMT3 can methylate the outer cytosine in CCGs only when the internal cytosine is methylated by MET1, whereas the methylation of CWGs only requires CMT3 [90]. (b) CMT3 is preferentially recruited to CCGs through minor SUVH5 and SUVH6 [89]. (c) CG and CCG methylations are rapidly restored during the S phase by MET1. In contrast, CWG methylation requires H3K9me2 deposition and must occur in G2 [84]. Thus, the methylation dynamics of rather pseudo symmetrical CCGs appear to be more like AT-rich CHHs than CWGs. 

The maintenance of CHH methylation is controlled through two pathways, DRM1/2 driven by RdDM [31] and CMT2, which can form a self-reinforcing loop with SUVH. However, the binding affinity is much lower than that of CMT3 [91], consistent with the negligible correlation between CHH methylation and rDNA silencing in *T. porrifolius*. The altered specificity of CMT2, preferentially targeting CAAs, CTAs, or CATs [89,92], corresponds well to the order observed in rDNA from *T. porrifolius*. The highly biased methylation pattern of CHHs, characterized by a small fraction (3%) of relatively highly methylated (>50%) AT-rich CHHs complemented by a relatively large fraction (30%) of poorly methylated (<1%) CHHs, may also be mediated by DRM2 directed to targets by specific ncRNAs. Such a biased methylation pattern, stably maintained in all rDNA variants regardless of transcriptional activity, indicates that the methylation patterns of AT-rich CHHs can also be guided by the surrounding sequences or chromatin structure. 

## 4. Material and Methods 

### 4.1. Plant Material and Drug Treatments

Two lineages of *T. porrifolius* were used in this work. The por1 lineage was obtained from Soltis and Soltis seed collection, 2611-2, originating from the Pullman region, Washington, the United States [43]. The por2 lineage originated from seed collection at Masaryk University, the Czech Republic (TP-1123-2001). It is necessary to emphasize that its 35S rDNA structural features were highly related to *T. porrifolius* accessions from Soltis and Soltis collection, 2612, originating from the Potlach region, Idaho, United States [43]. Obtained seeds were planted in a garden and flowering plants were repeatedly self-pollinated. Despite almost identical flower morphology (Appendix A), por1 plants were significantly taller than por2 plants (Appendix A), and the por1 lineage produced slightly more seeds on average than the por2 lineage (Appendix A).

Young leaves (whole centrally situated the youngest leaf weighing 80–120 mg), root tips (fresh 5–10 mm), and entire seedlings (five days old) were used for comparative transcription and methylation analyses. Leaves and root tips were harvested from five- to six-month-old plants (August, September); one por1 (por-34) and two por2 (por2-15 and por2-134) plants were used as material sources. Two samples of seedlings, one for each por1 and por2 lineage, were represented by five siblings that were combined and analyzed simultaneously. Germination conditions can be found in the following paragraph (control water). For FISH, meristematic cells were isolated from primary roots of three-day-old seedlings or roots of four-week-old plants grown in soil.

To induce DNA hypomethylation, por2 seeds (five for each experiment/drug) were treated with 200 µM of 5-azacytidine (AzaC) or 50 µM of 9-(2,3-dihydroxypropyl) adenine (DHPA) in water [93]. Histone acetylation was modified with 10 mM of sodium butyrate (NaBT) in water [44,94]. A mixture of 200 µM of AzaC with 10 mM of NaBT was also checked. The seedlings germinated in the water served as a control. During germination, the seeds were shaken on a horizontal shaker at 70 rpm, room temperature, and lighting conditions of 12 h light and 12 h dark. The solutions were replaced daily. Total DNAs and RNAs were isolated from each four- to five-day-old seedling (three biological replicates).

### 4.2. DNA and RNA Extractions and cDNA Preparations

Total DNAs were extracted using either the modified cetyltrimethylammonium bromide method [14] or a NucleoSpin Plant II Midi kit (Macherey-Nagel, Düren, Germany). Total RNAs were extracted using either a Tri Reagent (Sigma-Aldrich, St. Louis, MO, USA or RNeasy Plant Mini kit (QIAGEN, Hilden, Germany) and stripped of DNA traces using TURBO DNA-free (Invitrogen, Waltham, MA, USA). The quantity of nucleic acids was checked using a nanophotometer N60 (Implen GmbH, Munich, Germany) or fluorimeter Quantus (PROMEGA, Madison, WI, USA). The quality of nucleic acids was determined by agarose gel electrophoresis. Reverse transcriptions (20 µL), consisting of 1 µg of total RNA, 50 ng of random primers (Generi Biotech, Hradec Kralove, Czech Republic), 10 nmol of each dNTP, and 20 units of ProtoScript II (New England Biolabs, Ipswich, MA, USA), were followed by RNAse A treatments (Sigma-Aldrich, St. Louis, MO, USA).

### 4.3. PCR, CAPS, and Southern Hybridization

PCR amplification was used to distinguish between rDNA variants providing the PCR products of variable sizes (Figure 1b). In addition, PCR products of identical sizes were distinguished by Cleaved Amplified Polymorphic Sequences (CAPS) (Figure 2c,d). PCR amplification (20 µL) typically consisted of 3 ng of genomic DNA (0.4 µL of cDNA), 8 pmol of each primer (Appendix A), 4.8 nmol of each dNTP, and 0.8 units of Taq polymerase (Finnzymes, Espoo, Finland). If necessary, the PCR products were desalted using Qiex II (QIAGEN, Hilden, Germany). The 605 bp ETS2 region (Figure 2c) was amplified by primers ETS2F/ETS2R and cycling consisting of an initial denaturation step at 95 °C/180 s; 30 cycles of 95 °C/20 s, 60 °C/20 s, and 72 °C/40 s; and a final extension of 72 °C/5 min. After digestion with MboI, the DNA fragments were size separated by 2% agarose electrophoresis. Alternatively, PCR products were cloned by QIAGEN PCR Cloning plus Kit (QIAGEN, Hilden, Germany and Sanger-sequenced (Microsynth GmbH, Balgach, Switzerland) to evaluate sequence variability in ETS2 both in the genome and primary transcript. The 88 bp ETS2 region (Figure 2d) was amplified by primers ETS2F88/ETS2R88 and cycling consisting of an initial denaturation step at 95 °C/180 s; 30 cycles of 95 °C/20 s, 64 °C/20 s, and 72 °C/25 s; and a final extension of 72 °C/5 min. PCR products were digested with NlaIII and size-separated by 10% PAGE.

Southern hybridization was used to distinguish S- and L- rDNA variants based on the size variability of IGSs (Figure 1c). Genomic DNAs (1.5 µg) were digested with MboI (10 U; 2 × 4 h), size-separated by agarose electrophoresis, and blotted onto nylon membranes (GE Healthcare). The hybridization was carried out in a modified Church–Gilbert buffer [95]. The hybridization signals were visualized by Typhon FLA 9000 (GE HealthCare, Chicago, IL, USA) and quantitatively analyzed using ImageQuant (GE HealthCare, Chicago, IL, USA). The hybridization probe was represented by a mixture of 102–104 bp long R_103_ repeats derived from accessions FN666261 and HG915911. For preparation, an equimolar mixture of recombinant plasmids was used as a template for PCR with primes NTSF/ETS2R1 (Appendix A) and cycling consisted of an initial denaturation step at 95 °C/180 s; 30 cycles of 95 °C/20 s, 55 °C/20 s, and 72 °C/180 s; and a final 72 °C/5 min extension. PCR products were digested with BstNI, size-separated using 2% agarose electrophoresis, and purified with QiexII (QIAGEN, Hilden, Germany).

### 4.4. High-Throughput DNA Sequencing

About 5 μg of intact DNAs from each por1 and por2 were subjected to PacBio sequencing at SeqMe (Dobříš, Czech Republic) and GATC Biotech (Konstanz, Germany), respectively. The average lengths of PacBio reads were 5799 bp and 2901 bp for por1 and por2, respectively. The numbers of reads were 52,275 and 579,525 for por1 and por2, respectively. The read quality of each read was estimated from the number of corresponding circular subreads. Using CLC genomic Workbench 20.0 (QIAGEN, Hilden, Germany), 515 and 1042 PacBio reads from por1 and por2, respectively, were mapped to a reference sequence (9097 bp) assembled from available accessions (GenBank) for the *Tragopogon* 26S rRNA gene (AF036493.1 and KT179725.1 from *T. dubius*), 18S rRNA gene (KT179662 and U42502 from *T. dubius*), IGS (HG915911 from *T. porrifolius*), and ITS1-5.8S_rDNA_gene–ITS2 region (consensus sequence derived from *T. porrifolius* accessions FN675706, FN675707, FN675708, FN675709, FN675710, and AJ633494). Because most mapped reads contained substantially longer IGS than the original reference sequence, a new, more representative reference sequence (11,265 bp) was constructed from all mapped reads. Remapping all accessible reads to this longer reference yielded 621 and 1150 mapped reads from por1 and por2, respectively. Newly mapped reads were mostly homologous to highly variable repetitive IGS, suggesting the legitimacy of repeated mapping at complex repetitive regions, particularly in rDNA IGS.

For RNA sequencing, 5 µg of DNA-free, high-quality total RNAs was sequenced at SeqMe (Dobříš, Czech Republic) using the Illumina platform. To estimate the proportions of individual rRNA variants in primary transcripts, Illumina reads were mapped by CLC genomic Workbench 20.0 (QIAGEN, Hilden, Germany) to several reference sequences covering variable sites distinguishing individual rDNA variants.

For cytosine methylation analysis, 3–5 μg of high-quality DNAs were bisulfite-treated and Illumina-sequenced at Macrogen, Inc. (Seoul, Republic of Korea). The genome coverage, read quality, and other parameters are listed in Appendix A. Using the CLC genomic Workbench 20.0 (QIAGEN, Hilden, Germany), Illumina reads were mapped to several c. 120 nt long reference sequences within ETS2 (regions I–VI; consensus sequence derived from long PacBio reads), which distinguished rDNA variants distinct in specific transcriptional activity. Mapped reads were aligned using the CLC genomic Workbench 20.0 (QIAGEN, Hilden, Germany), and only reads identical to the reference sequence were subjected to methylation analysis using CyMate (Cytosine Methylation Analysis Tool for Everyone; http://www.cymate.org; access from 4 October 2022 to 30 February 2024) (96). To achieve well-supported estimations of the differences between the methylation of individual rDNA variants, we used both basic tools provided by this program: (i) the average methylation level of each respective cytosine and (ii) the proportion of Illumina reads with all demethylated CWGs or CCGs or CAAs or CATs. For highly methylated CGs, this mode was used to estimate the proportions of reads where at least one of the neighboring CGs was demethylated.

### 4.5. Sequence Analyses

The copy numbers and lengths of tandemly arrayed monomeric units (R_103_, R_156_, R_7_), as well as the overall lengths of repetitive regions in IGSs, were estimated using the Tandem Repeats Finder (97) (https://tandem.bu.edu/trf/trf.html; access from 15 January 2022 to 22 February 2022) and dot plot diagrams constructed by either the YASS genomic similarity search tool (http://bioinfo.unif-lille.fr/yass/; access from 17 January 2022 to 20 February 2022) or dotmatcher (http://www.bioinformatics.nl/cgi-bin/emboss/dotmatcher; access from 25 January 2022 to 26 February 2022).

The ETS2 sequences were aligned either using the MUSCLE alignment software implemented in MEGA7 Molecular Evolutionary Genetics Analysis 10.1.1. (long Sanger or PacBio; Figure 1d) or using BioEdit Sequence Alignment Editor 7.2.5 (12/11/2013) (short Illumina—methylation analyses), and the phylogenetic trees were constructed with the MEGA 7 [96]. The evolutionary history was inferred using a maximum likelihood method based on the Tamura–Nei model (98). The confidence of inferred evolutionary relationships was assessed using bootstrap analysis with 1000 repetitions. The sequence identity between individual promoters was computed from the SNP distributions using the Sequence Identity matrix implemented in BioEdit 7.2.5 (12/11/2013) (Appendix A).

Statistical supports of the differences between samples (methylation density between rDNA variants, between tissues, and between individual cytosines) were estimated by both the non-parametric Mann–Whitney U-test (http://vassarstats.net/utest.html; access from 11 March 2022 to 13 February 2024) and parametric Student’s *t*-test (Microsoft Excel, Microsoft 365). The difference “D_s-a_” between methylation medians (means) of silent (lower specific transcriptional activity) rDNA variant and active (higher specific transcriptional activity) rDNA was used for semiquantitative evaluations of absolute differences [%] and directionality in the methylation of each pair of rDNA variants.

### 4.6. Fluorescence In Situ Hybridization

Chromosome squashes were prepared according to [97], with slight modifications. Enzymatic treatment of roots prior to squashing lasted 90 min. Slides were treated with RNase A (100 µg/mL) in a humidified chamber at 37 °C for 1 h, rinsed in 2 × standard saline citrate (SSC) at room temperature (RT) for 3 × 5 min, treated with pepsin (50 µg/mL) at RT for 5 min, rinsed in 2 × SSC (3 × 5 min), fixed in 3.7% formaldehyde/phosphate-buffered saline (PBS) for 10 min, rinsed in 2 × SSC (3 × 5 min), dehydrated in an ethanol series (75%, 90%, and 100%) for 2 min each, and air-dried. The probes used were 18S rDNA (a cloned gene from a tomato, GenBank # X51576.1) and 5S rDNA (an *Artemisia tridentata* S4 clone, GenBank # JX101915.1). The 5S probe was labeled with SpectrumGreen dUTP using a Nick Translation Kit (Abbott Molecular, Des Plaines, IL, USA). The 18S probe was labeled with Amersham FluoroLinkTM Cy3-dUTP (GE Healthcare, Chalfont Saint Giles, England) using the Nick Translation Mix from Roche (Basel, Switzerland) according to the manufacturer’s instructions. The hybridization mix (50% formamide, 10% dextran sulfate, 0.2 × SSC, and 75–100 ng of each labeled probe in a 30 µL volume) was heated (10 min/75 °C), cooled on ice (5 min), pipetted onto a slide, and covered with a plastic coverslip. The slide was incubated in a humidified chamber at 75 °C/5 min, 65 °C/2 min, 55 °C/2 min, 45 °C/2 min, and 37 °C for 35 hrs and then washed for 2 × 5 min in 2 × SSC at 42 °C, 2 × 5 min in 0.1 × SSC at 42 °C, 2 × 5 min in 2 × SSC at 42 °C, 5 min in 2 × SSC at RT, and 7 min in 4 × SSC/0.1% Tween 20 at RT, with final brief wash in PBS. The chromosomes were counterstained in Vectashield (Vector Laboratories, Inc., Burlingame, CA, USA) containing 1.5 µg/mL of 4′,6-diamidino-2′-phenylindole (DAPI). Fluorescence images were captured using an Olympus AX 70 fluorescence microscope equipped with a digital camera. Images were analyzed and processed using ISIS software (MetaSystems, Altlussheim, Germany). Chosen chromosomes were cut from the picture in Adobe Photoshop CS6.

### 4.7. Data Availability

The FASTQ sequencing data generated from this study were submitted to NCBI as twelve biosamples (Appendix A) deposited in the BioProject ID PRJNA634996. The FASTA sequencing data generated from Sanger sequencing of ETS2 cDNA were submitted to NCBI under GenBank accessions MW036554–MW036631.

## 5. Conclusions

Relationships between transcription and cytosine methylation in individual sequence contexts were comprehensively characterized in several genetically highly related 35S rDNA variants, native to two *T. porrifolius* lineages substantially differing in rDNA copy numbers and the direction of nucleolar dominance. The methylation dynamics of CG, CWG, CCG, and CHH motifs contributed differently to global methylation changes associated with transcriptional silencing/activation, suggesting that they can be differentially involved in the regulation of rDNA transcription in *T. porrifolius*. The most relevant is CWG methylation as follows from the statistically best-supported correlations between silencing and cytosine methylation in tissues with a high mitotic index, such as root tips and seedlings, but not in young leaves. A self-reinforcing loop between CWG methylation and H3K9 methylation may be responsible for the most prominent epigenetic changes in chromatin structure leading to complete but reversible silencing of rDNA in *T. porrifolius*. In contrast, highly stable methylations in individual rDNA variants, as well as in individual tissues, were associated with the external cytosine in CCG motifs and some AT-rich CHH motifs, whose methylations were rather position-dependent. The CCG methylation dynamics appeared to be more reminiscent of CAAs (CHHs) than CWGs, as evidenced by the wide variability of methylations between individual CCGs and no correlations between methylation and silencing.

## Figures and Tables

**Figure 1 ijms-25-07540-f001:**
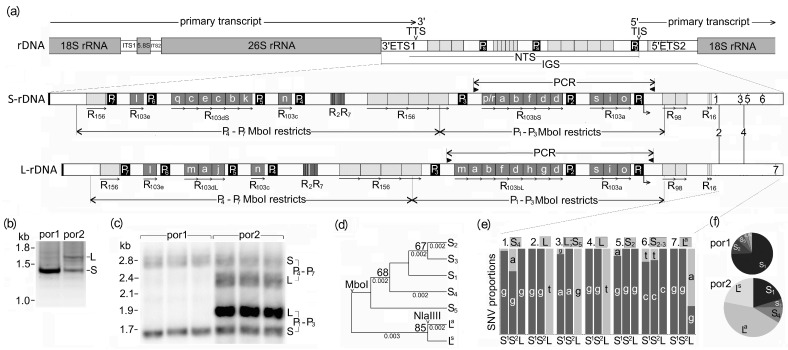
Structural differences between rDNA variants in *T. porrifolius.* (**a**) The common structure of the plant 35S rDNA repeat unit and the arrangements of seven PolI promoter–like elements (P) and various tandem repeats (R) in the two most prominent S- and L-rDNA variants in T. porrifolius. Fine variants of R_103_ repeats are distinguished by the letters a–s. Seven abundant SNPs found within 5′ETS2 are numbered 1–7. Four (1, 3, 5, and 6) are specific for S-rDNA, one (7) is specific for L-rDNA, and two (2 and 4) distinguish L- and S-rDNA variants. Locations and expected sizes of PCR products and Mbol-digested fragments are highlighted. (**b**) PCR products using primers NTSF and ETS2R1 (Appendix A). (**c**) Southern hybridizations of the R_103_ probe to MboI-restricted genomic DNAs. Because MboI targets are absent from each R_103_ and P repeat but present in each ETS1, ETS2, and R_156_ region, complete P_1_–P_3_ and P_4_–P_7_ regions are released as long blocks of sizes specific for S- and L-rDNA variants. The structural uniformity of IGSs within the por1 lineage was documented by two prominent hybridization signals detected in each individual, whereas four signals were characteristic for each por2 plant. (**d**) Based on seven SNPs within ETS2 (panel **a**), the maximum likelihood method and the Tamura–Nei model revealed three statistically well-supported rDNA branches (S_1-5_, L^a^, and L^s^), which correlate with the occurrence of targets for NlaIII and MboI. (**e**) The proportions of individual nucleotides participating in seven abundant SNVs within ETS2 from por1 (columns S^1^) and por2 (columns S^2^ and L). The assignments to selected rDNA variants are shown above the columns. (**f**) The overall proportions of individual rDNA variants in por1 and por2 genomes were estimated from (panel **f**).

**Figure 2 ijms-25-07540-f002:**
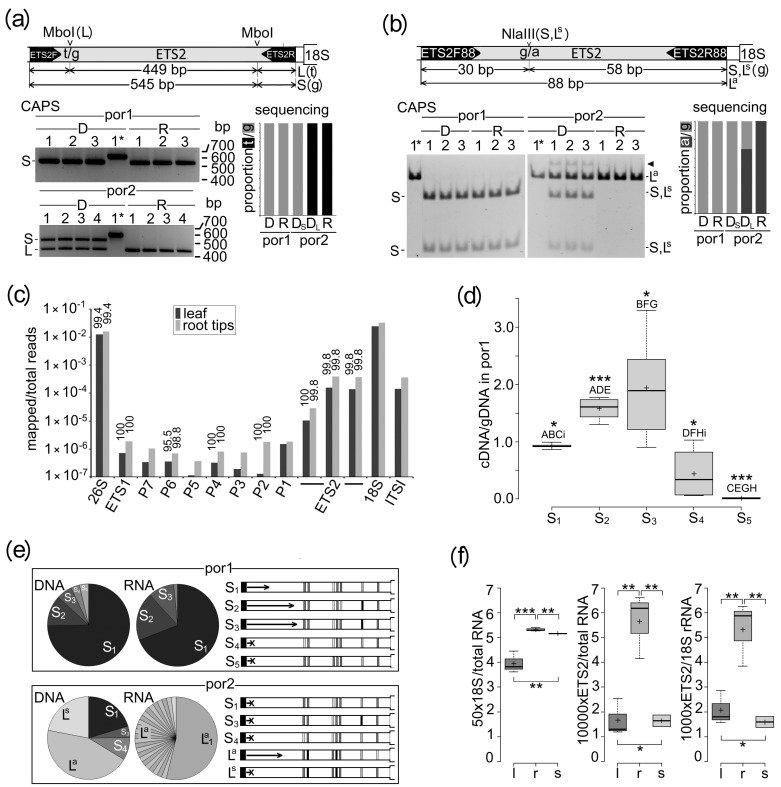
Transcription of individual rDNA variants in por1 and por2. (**a**,**b**) CAPS analyses of indicated restriction targets which distinguished either L-rDNA from S-rDNA (**a**) or L^a^-rDNA from both L^s^- and S-rDNAs (**b**) in genomes (columns D) and primary transcripts (R). DNAs were separated either in 1.5% agarose (**a**) or in 10% polyacrylamide (**b**) gels. Asterisks indicate lanes with undigested PCR products as control. Arrowheads indicate the hypothetical DNA-NlaIII adduct. Bar charts show the proportions of diagnostic targets in genomes (columns D_S_ and D_L_) and primary transcripts (R) detected by sequencing. (**c**) The number of Illumina reads derived from transcriptome and mapped to individual rDNA regions related to the total number of reads in both leaves and root tips. The percentage of the L-rDNA is shown above each relevant column. (**d**) The transcriptional efficiencies (specific transcription) of individual S-rDNA variants in por1 are demonstrated as the ratio between the proportions of a given variant in the primary transcript (cDNA) and genome (gDNA) (Appendix A). Statistically significantly supported (*p2*< 0.01) differences between variants are highlighted by identical capital letters; moderately (0.01 < *p2* < 0.05) supported differences are highlighted by identical small letters. One asterisk indicates statistically unsupported (*p2* > 0.05) differences between proportions of a given rDNA variant in genomes and transcriptomes; three asterisks indicate significant (*p2* < 0.01) differences between proportions of a given rDNA variant in genomes and transcriptomes. (**e**) Proportions of abundant rDNA variants in genomes (DNA) and transcriptomes (RNA) from por1 and por2 (pie charts) indicate differing transcriptional efficiency (arrow lengths) of individual rDNA variants, differentiated by seven polymorphic sites in ETS2. (**f**) Relative contents of ETS2 and 18S rRNA transcripts in leaves (l), root tips (r), and seedlings (s). Mapped ETS2 transcripts were related either to mature 18S rRNAs or total RNAs. In addition, mature 18S rRNAs were related to total RNAs (Appendix A). Box plots represent the minimum, 1st quartile, median, 3rd quartile, maximum, and mean (+). One, two, and three asterisks indicate statistically unsupported (*p2* > 0.05), moderately supported (0.01 < *p2* < 0.05), and significantly supported (*p2* < 0.01) differences, respectively.

**Figure 3 ijms-25-07540-f003:**
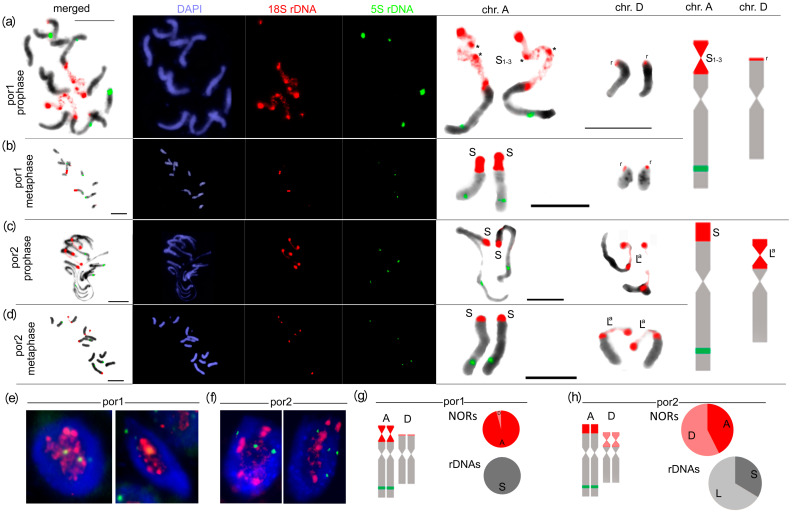
Secondary constrictions in por1 can be assigned to A-NORs, whereas secondary constrictions in por2 are associated with D-NORs. FISH of 18S rDNA (red) and 5S rDNA (green) to prophases (**a**,**c**), metaphases (**b**,**d**), and interphases (**e**,**f**) in por1 (**a**,**b**,**e**) and por2 (**c**,**d**,**f**). Presumptive chromosomal localizations of individual rDNA variants are depicted on excised and enlarged chromosomes A and D on the right side of each panel (**a**–**d**). In por1, the decondensed 35S rDNA region on each chromosome A is split by two condensed foci (asterisk) into three decondensed subregions, which may correspond to three transcribed S_1–3_-rDNA variants. The decondensed 35S rDNA region on each chromosome D is uninterrupted and perhaps corresponds to a single abundant transcribed L^a^-rDNA variant detected in por2. 35S and 5S signals were assigned to individual chromosomes according to the Ownbey and McCollumns nomenclature [41,43]. Chromosomes were counterstained with DAPI (blue). Representatives were selected from 34 analyzed metaphases (prophases) for each por1 and por2. The relative sizes of 35S rDNA signals associated with A- and D-NORs were averaged from seven metaphases and presented as pie charts for both por1 (**g**) and por2 (**h**) lineages. For comparison, the proportions of S- and L-rDNA variants are shown in the same way.

**Figure 4 ijms-25-07540-f004:**
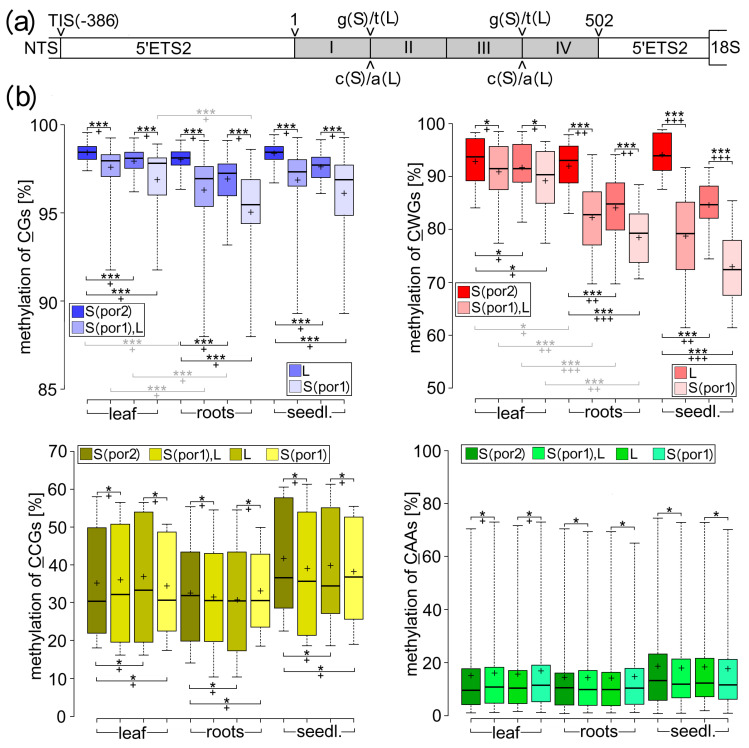
Methylation dynamics of individual C-motifs within L- and S-rDNAs. (**a**) Two g>t (c>a) substitutions distinguished bisulfite-modified L- and S-rDNAs within ETS2 (regions I–IV). (**b**) Methylation ranges of indicated C-motifs for a given tissue were compared between (i) silent S-rDNA from por2 and all active S-rDNA from por1 and L-rDNA from por2, (ii) L-rDNA from por2 and S-rDNA from por1, (iii) S-rDNA and L-rDNA from por2, and (iv) S-rDNA from por2 and por1. In addition, selected pairwise comparisons between leaves and root tips were performed for a given rDNA variant (gray). All comparisons were evaluated in Appendix A, sheet 4A. Box plots are constructed using the minimum, 1st quartile, median, 3rd quartile, maximum, and mean (+). One, and three asterisks indicate statistically unsupported (*p2* > 0.05), and significantly supported (*p2* < 0.01) differences, respectively, between compared samples. One, two, and three crosses indicate |D_s-a_| < 5%, 5% < |D_s-a_| < 10% and 10% < |D_s-a_|, respectively (Materials and Methods). Differences between rDNA variants are highlighted in black, whereas differences between tissues are gray. Complementary analyses showing the proportions of hypomethylated Illumina reads and the differences between L^a^ and (S + L^s^) rDNA variants are shown in Appendix A.

**Figure 5 ijms-25-07540-f005:**
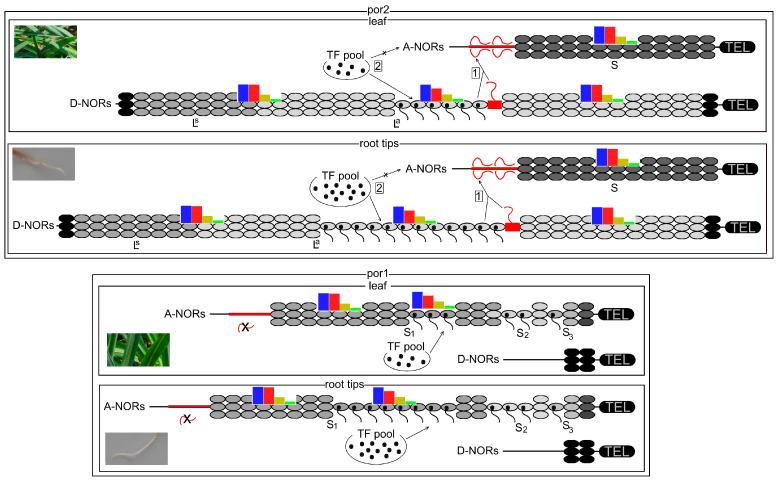
Proposed mechanisms participating in the regulation of ND in *T. porrifolius*. (1) Transcriptionally active D-NORs in por2 mediate the silencing of A-NORs through some *in trans* acting factors. (2) Some limited TFs are exhausted from A-NORs due to their elevated affinity to D-NORs. Higher rDNA transcription in root tips can be due to a higher content of TF. The action of *in trans* acting factors and lack of TF can induce locus-specific chromatin condensation and transcriptional silencing. Deleting a large part of both D-NORs can lead to the extinction of *in trans* acting factors and the availability of all TFs, followed by chromatin decondensation and transcriptional activation of A-NORs in por1. Condensed and decondensed chromatin can differ in DNA methylation, mainly in CWGs (red bars) and, to a lower extent, in CGs (blue). In contrast, methylation of CCG (yellow) and CHH (green) motifs are invariable. Because the proportions of both chromatin conformations can significantly differ between NORs as well as between tissues, overall methylation patterns in individual rDNA variants and tissues depend on these proportions. Clustering, rather than intermingling, and relative spatial position along NORs may affect the specific transcriptional activity of individual rDNA variants within A-NOR (S_1–5_) and D-NOR (L^a^ and L^s^). Only one NOR is shown for each A- and D-pair, and the length of constriction is proportional to the transcriptional activity.

## Data Availability

Data are contained within the article and Appendix A.

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
