# Peer review of "Transcriptional Silencing of 35S rDNA in Tragopogon porrifolius Correlates with Cytosine Methylation in Sequence-Specific Manner"

_ijms, 2024, doi:10.3390/ijms25147540_

Round 1
Reviewer 1 Report
Comments and Suggestions for Authors
Authors studied genetic and epigenetic relationships between A- NORs and D- NORs in T. porrifolius using global repetitive organization of IGS, transcriptional activity chromatin structure and cytosine methylation sequencing in both A- and D-NORs. I appreciate the large amount of work and efforts. However, there are many concerns that needs to be address. English writing style is horrible. its messy and hard to understand. Some of the sentences has no meaning and makes no sense. I give here one example but there are many sentences like that:
"Three biological replicates of both leaves and root tips have originated from five to six months old por1-34, por2-134 and por2-15 plants." what does exactly this sentence mean?
Abstract:
It's written like introduction and discussion. I couldn't see any clue about hypothesis and the experimental method in abstract. This part definitely needs to be re-written.
Introduction:
This part is too long. needs to be shortened. Moreover, sentences are too long. For example from line 119 to 122 is one sentences which is really unclear and confusing.
Results
As I mentioned before. The English writing style is confusing and hard to understand. Why Table 1 is presented in the middle of introduction? Tables are messy. You almost put half of the alphabet in table 2. It is hard to fallow up the text and tables. Please provide figures more separately.
M&M
Definitely needs to be re written. specially section 5.1 and 5.3
Line 801-802: Its 35S rDNA structural complexity is highly related to..... what does that means?
5.3. PCR, CAPS and Southern hybridization. PCR of what? Which gene? You performed PCR to check what? All needs to be explained. You provided table A4 as primers that used for PCR. I can't see explanation in tables captions that from where you got that primers or those are related to which gene? Another unusual thing is why all tables are name with "A"? If they are provided as supplementary tables why not "S" and "A"?
Comments on the Quality of English LanguageDear Editor
I appreciate the idea and effort. Manuscript has potential for publication. However it is written very messy and English language is hard to understand. Unfortunately, it is not possible to evaluate it scientifically due to the writing and language errors. Authors may can submit the revised verssion.
Best regards
Mortaza Khodaeiaminjan
Author Response
Dear Reviewer,
thank you very much for taking the time to review this manuscript. Please find the detailed responses below and the corresponding revisions/corrections highlighted/in track changes in the re-submitted files.
Authors studied genetic and epigenetic relationships between A- NORs and D- NORs in T. porrifolius using global repetitive organization of IGS, transcriptional activity chromatin structure and cytosine methylation sequencing in both A- and D-NORs. I appreciate the large amount of work and efforts. However, there are many concerns that needs to be address. English writing style is horrible. its messy and hard to understand. Some of the sentences has no meaning and makes no sense. I give here one example but there are many sentences like that:
Response 1.Thank you for this valid criticism. Indeed, the article was not revised by a native speaker. Thus, comprehensive language proofreading was carried out using the MDPI service.
"Three biological replicates of both leaves and root tips have originated from five to six months old por1-34, por2-134 and por2-15 plants." what does exactly this sentence mean?
Response2. Thank you, I agree that this sentence was written very sparingly. The plant material used in the work is currently described in more detail.
Abstract:
It's written like introduction and discussion. I couldn't see any clue about hypothesis and the experimental method in abstract. This part definitely needs to be re-written.
Response 3. Thank you. The abstract was completely re-written taking into account the used methodology summarized at the beginning and the hypothesis formulated at the end.
Introduction:
This part is too long. needs to be shortened. Moreover, sentences are too long. For example from line 119 to 122 is one sentences which is really unclear and confusing.
Response 4. The introduction was shortened extensively. Overly long and often unclear sentences were either reformulated or shortened throughout the manuscript
Results
As I mentioned before. The English writing style is confusing and hard to understand. Why Table 1 is presented in the middle of introduction? Tables are messy. You almost put half of the alphabet in table 2. It is hard to fallow up the text and tables. Please provide figures more separately.
Response 5. Language proofreading was carried out using the MDPI service. It is true that some tables were too complicated and sometimes showed too much data, thank you for this comment. Table S2 was completely reorganized. Table S8 has been simplified in that only its second part is now presented.
M&M
Definitely needs to be re written. specially section 5.1 and 5.3
Response 6. It was done, thank you
Line 801-802: Its 35S rDNA structural complexity is highly related to..... what does that means?
Response 7. Thank you, it was corrected
5.3. PCR, CAPS and Southern hybridization. PCR of what? Which gene? You performed PCR to check what? All needs to be explained. You provided table A4 as primers that used for PCR. I can't see explanation in tables captions that from where you got that primers or those are related to which gene? Another unusual thing is why all tables are name with "A"? If they are provided as supplementary tables why not "S" and "A"?
Response 8. Thank you, the application of PCR, CAPS and Southern was explained in the appropriate places in the manuscript, mainly in the Material and Methods. Table S4 was extensively reorganized. I apologize for confusing A and S in the picture and table captions. Now S is used everywhere.
Comments on the Quality of English Language
Response 9. Language proofreading was carried out using the MDPI service
Reviewer 2 Report
Comments and Suggestions for Authors
This is an interesting paper analysing in detail the configuration and methylation of rDNA variants in Tragopogon porrifoIius. A combination of DNA sequencing, RNA sequencing, PCR, Southern hybridization and FISH was used to correlate RNA variants, expression and DNA methylation.
My concerns are:
1.) I found the manuscript in place difficult to follow, maybe due to the complex nature of the experiments, but also due to some inconsistency in naming the variants and how they relate from the outset of the paper: S-rDNA and L-rDNA variants versus S-IGS and L-IGS variants, S-ETS2 and L-ETS2, , A-NOR and D-NOR. Variants Por1 and Por2 are not explained in the abstract, introduction or Figure 1 legend.
2.) My main concern of the paper is that the evidence for A- and D- NORs being S-rDNA and L-rDNA is circumstantial by correlation of size and abundance of IGS variants (lines 357ff) - is this real? I agree that methylation patterns can be correlated to sequence variants by SNPs, but how do you know which sequence or variant is located on which chromosome?
3.) Most of the discussion concentrates on a model of silencing and activation of the rDNA (Figure 5), but much is speculation; although I do not want to discourage this all together, I feel large sections of the discussion go far beyond the experimental evidence that is provided by the paper, and I think should be shortened.
4.) I found the abstract poor and in places difficult to what is written in main text; mainly it is too generic and does not give detail about the variants analysed. And in line 20 ‘So, new S-rDNA epigenetic variants arose in A-NORs following disruption of interactions between NORs, probably controlled by intact D-NORs.’ Does not really follow from the previous sentence.
5.) In the result sections, various interpretations, conclusion and discussion points are made without giving evidence or references. Some are the end of sections. While it is good to have interpretations and a short summary or take home message before going on, assumptions and interpretations should not be made without support, and the reader should not think ‘why’. I noted in particular:
Line 199/200: ‘We suggest that minor IGS variants may represent either nonfunctional pseudogenes or they may increase plasticity in transcriptional regulation depending on the need for proteosynthesis.‘
Lines 301-307 ’Collectively, distinct epigenetic variants may be associated with each S- and L- rDNA as follows from the occurrence of: (i) active and silent S1 -ETS2 in por1 and por2, respectively, (ii) active (La) and silent (Ls) variants in por2 and (iii) five ETS2 variants with different specific transcription efficiency in por1. As all these variants are genetically almost identical (common R103bS satellite, ETS2, ETS1), we hypothesize that they could differ mainly in higher order chromatin structure, distribution of specific epigenetic markers and perhaps also in spatial localization within rDNA arrays’
Lines 328-339: ‘The tiny rDNA arrays, that likely remained on both D-chromosomes in por1 after homologous deletion of a large portion of the transcriptionally active rDNA units, may be formed from low-copy rDNA variants, possibly pseudogenes, detected in both por1 and por2 lines (Figure A1b,c). The chromosomal break probably occurred at the interface between heterochromatin and euchromatin. Relative sizes of individual rDNA loci and their epigenetic (condensed/decondensed) patterns were similar in all analyzed individuals within each por1 and por2 lineage (Figures 3 and A3). Such cytogenetic uniformity and differences found within and between both lineages, respectively, well corresponded to distinct structural (Figure 1) and transcriptional (Figure 2) features of 35S rDNA in por1 and por2. Collectively, reciprocal nucleolar dominance associated either with A-NORs in por1 or with D-NORs in por2 occurred in T. porrifolius. ‘
Lines 492-494: ‘These results indicate that methylation patterns of AT-rich CHHs and CCGs can be guided by surrounding sequences or chromatin structure independently of transcriptional activity.’ - I particularly wonder here, how you know this?
Lines 497-508: Whole paragraph feels more a discussion
6) Some minor points
Lines 377 - 379: include how you measured transcription.
Lines 383/384: what are the morphological changes
Section 2.3: I suggest to describe por1 first and then por2 as is depicted in Figure 3.
7.) Beginning of the discussion is confusing as information is missing
Line 532: flower morphology not mentioned before but later in M&M
Line 533: two lineages, name them here for the benefit of the reader
Line 538: should the ‘and’ be an ‘or’?
Line 538/539: ‘ETS2 sequences did not provide evidence that the por1 and por2 arose by hybridization between progenitors with the last two karyotypes.’ - This sentence is not clear; no mention before that por1 and por2 have A and D NORs.
Line 540, add ‘if’ before inherited to indicate the speculative assumption you are making.
Comments on the Quality of English Language
Overall, I think the MS will need a careful checking of English syntax and organisation; there are the odd words missing or extra, as well as sometimes awkward and convoluted constructions including those that make the meaning unclear. e.g. the use of ‘respectively’ where it would be better to mention each scenario in full before the next; see Line 163: ‘detected one (S) and two (S and L) prominent IGS variants in por1 and por2 lineages, respectively’; I think this needs and ‘OR’ when using respectively; and probably you mean one (S) in por1 and/or two (S and L) in por2; please make it clear. Delete the ‘respectively’ in line 302 and say active in por1 and inactive in por2; in line 324, line 359, line 504 and others.
Author Response
Dear Reviewer,
thank you very much for taking the time to review this manuscript. Please find the detailed responses below and the corresponding revisions/corrections highlighted/in track changes in the re-submitted files.
This is an interesting paper analysing in detail the configuration and methylation of rDNA variants in Tragopogon porrifoIius. A combination of DNA sequencing, RNA sequencing, PCR, Southern hybridization and FISH was used to correlate RNA variants, expression and DNA methylation.
My concerns are:
1.) I found the manuscript in place difficult to follow, maybe due to the complex nature of the experiments, but also due to some inconsistency in naming the variants and how they relate from the outset of the paper: S-rDNA and L-rDNA variants versus S-IGS and L-IGS variants, S-ETS2 and L-ETS2, , A-NOR and D-NOR. Variants Por1 and Por2 are not explained in the abstract, introduction or Figure 1 legend.
Response 1.Thank you for pointing out the overly complex nomenclature of rDNA variants. In revised manuscript, we strictly used S-rDNA and L-rDNA…. variants with eventual additional specifications for individual regions as are IGS or ETS. Por1 and por2 lineages are now defined in Abstract and Introduction and comprehensively described later in Material and Methods section.
2.) My main concern of the paper is that the evidence for A- and D- NORs being S-rDNA and L-rDNA is circumstantial by correlation of size and abundance of IGS variants (lines 357ff) - is this real? I agree that methylation patterns can be correlated to sequence variants by SNPs, but how do you know which sequence or variant is located on which chromosome?
Response 2. Owing to extraordinarily similar nucleotide sequences of S-rDNA and L-rDNA variants, even within the IGSs, we were unable to design specific probes for direct assignment of individual rDNA variants to corresponding NORs by FISH. This fact is newly emphasized at the appropriate place in the manuscript. Therefore, we decided to perform these assignments indirectly by comparisons of por1 and por2 lineages distinct in number of rDNA variants and NORs. So, the assignment of S-rDNA structural variants to A-NORs and L-rDNA variants to D-NORs was indeed made based on the correlation between the numbers of major rDNA structural variants and the numbers of comparably sized NORs between Por1 and Por2 lineages. In addition these assignments were also supported by correlation between (i) L-rDNA transcriptional dominance and decondensation of D-NORs and (ii) S-rDNA silencing and decondensation of A-NORs. Thank you.
3.) Most of the discussion concentrates on a model of silencing and activation of the rDNA (Figure 5), but much is speculation; although I do not want to discourage this all together, I feel large sections of the discussion go far beyond the experimental evidence that is provided by the paper, and I think should be shortened.
Response 3. I agree, most rather speculations were deleted and highlighted. Thank you
4.) I found the abstract poor and in places difficult to what is written in main text; mainly it is too generic and does not give detail about the variants analysed. And in line 20 ‘So, new S-rDNA epigenetic variants arose in A-NORs following disruption of interactions between NORs, probably controlled by intact D-NORs.’ Does not really follow from the previous sentence.
Response 4. Thank you. The abstract was thoroughly revised and re-written.
5.) In the result sections, various interpretations, conclusion and discussion points are made without giving evidence or references. Some are the end of sections. While it is good to have interpretations and a short summary or take home message before going on, assumptions and interpretations should not be made without support, and the reader should not think ‘why’. I noted in particular:
Line 199/200: ‘We suggest that minor IGS variants may represent either nonfunctional pseudogenes or they may increase plasticity in transcriptional regulation depending on the need for proteosynthesis.‘
Response 5. The sentence has been deleted, thank you.
Lines 301-307 ’Collectively, distinct epigenetic variants may be associated with each S- and L- rDNA as follows from the occurrence of: (i) active and silent S1 -ETS2 in por1 and por2, respectively, (ii) active (La) and silent (Ls) variants in por2 and (iii) five ETS2 variants with different specific transcription efficiency in por1. As all these variants are genetically almost identical (common R103bS satellite, ETS2, ETS1), we hypothesize that they could differ mainly in higher order chromatin structure, distribution of specific epigenetic markers and perhaps also in spatial localization within rDNA arrays’
Response 6. The paragraph has been removed from the section Results and relevant ideas have been transferred to the section Discussion (highlighted). Thank you.
Lines 328-339: ‘The tiny rDNA arrays, that likely remained on both D-chromosomes in por1 after homologous deletion of a large portion of the transcriptionally active rDNA units, may be formed from low-copy rDNA variants, possibly pseudogenes, detected in both por1 and por2 lines (Figure A1b,c). The chromosomal break probably occurred at the interface between heterochromatin and euchromatin. Relative sizes of individual rDNA loci and their epigenetic (condensed/decondensed) patterns were similar in all analyzed individuals within each por1 and por2 lineage (Figures 3 and A3). Such cytogenetic uniformity and differences found within and between both lineages, respectively, well corresponded to distinct structural (Figure 1) and transcriptional (Figure 2) features of 35S rDNA in por1 and por2. Collectively, reciprocal nucleolar dominance associated either with A-NORs in por1 or with D-NORs in por2 occurred in T. porrifolius. ‘
Response 7. The first two sentences have been removed because they are really not supported directly by our experiments. The remaining part has been left in a somewhat modified wording as it directly relates to the results: (i) it emphasizes that the cytogenetic results documented in the Figures 3 and S3 were comparable in all analysed samples (plants) - statistical support. (ii) It highlights correlations between cytogenetic (FISH) and molecular (RNA, DNA sequencing, Southern, CAPS) results from previous paragraphs. (ii) rDNA transcriptional pattern in por1 and por2 lineages demonstrate bidirectional nucleolar dominance in T. porrifolius.
Lines 492-494: ‘These results indicate that methylation patterns of AT-rich CHHs and CCGs can be guided by surrounding sequences or chromatin structure independently of transcriptional activity.’ - I particularly wonder here, how you know this?
Response 8. This sentence was modified and transferred into Discussion
Lines 497-508: Whole paragraph feels more a discussion
Response 9. This paragraph was modified and relocated into section Discussion
6) Some minor points
Lines 377 - 379: include how you measured transcription.
Response 10. Relative transcription of S- and L- rDNAs was measured by RT_CAPS. Inserted in the text at the appropriate place
Lines 383/384: what are the morphological changes
Response 11. Similar morphological changes are represented mainly by delayed growth of primary root – added to text
Section 2.3: I suggest to describe por1 first and then por2 as is depicted in Figure 3.
Response 12. Thank you, an exchange was made
7.) Beginning of the discussion is confusing as information is missing
Line 532: flower morphology not mentioned before but later in M&M
Response 13. Thank you, the link to the corresponding Figure S10, as well as link to Figures 3 and S3 have been added
Line 533: two lineages, name them here for the benefit of the reader
Response 14. It was done, thank you.
Line 538: should the ‘and’ be an ‘or’?
Response 15. Yes, this can be better. It was changed. Thank you.
Line 538/539: ‘ETS2 sequences did not provide evidence that the por1 and por2 arose by hybridization between progenitors with the last two karyotypes.’ - This sentence is not clear; no mention before that por1 and por2 have A and D NORs.
Response 16. As follows from Figure 3, both por1 and por2 have A-NORs on chromosomes A and D-NORs on chromosomes D, although both D-NORs in por1 are negligible. Nevertheless this is newly emphasized in the beginning of Discussion. In addition this sentence is slightly modified for better clarify.
Line 540, add ‘if’ before inherited to indicate the speculative assumption you are making.
Response 17. It was done. Thank you for the advice
Comments on the Quality of English Language
Overall, I think the MS will need a careful checking of English syntax and organisation; there are the odd words missing or extra, as well as sometimes awkward and convoluted constructions including those that make the meaning unclear. e.g. the use of ‘respectively’ where it would be better to mention each scenario in full before the next; see Line 163: ‘detected one (S) and two (S and L) prominent IGS variants in por1 and por2 lineages, respectively’; I think this needs and ‘OR’ when using respectively; and probably you mean one (S) in por1 and/or two (S and L) in por2; please make it clear. Delete the ‘respectively’ in line 302 and say active in por1 and inactive in por2; in line 324, line 359, line 504 and others.
Response 18. Thank you. Language proofreading was carried out using the MDPI service. All unclear sites with „respectively“ were approppriately modified.
Round 2
Reviewer 1 Report
Comments and Suggestions for Authors
Dear Authors
I appreciate your effort and idea. Your work have good potential but unfortunately your writing style has still big problems. Please note that the English editing services only check the grammar and they can not improve your text content, writing and presenting style. I give you some exmple:
Line 37: which, may, however. How can you use all 3 together???
You gave the Figure 1 in line 51. However, until line 105 you don’t talk about Tragopogon porrifolius at all. Re organize your introduction. Figure 1b is coming from results and what a figure from results is doing in introduction part?
What does this sentence mean from line 128???
Long PacBio sequences showed that the IGSs in both por1 and popr2 lineages are built of common sequence motifs mainly represented by multiple kinds of repetitive DNA sequences, both tandemly (R) and dispersedly (P-promoters) arranged (Figure 1b).
Another example is from Line 153 to 156
Most of the core promoters (P1, P2, P3, P6, and P7) are identical in both L- and S-rDNAs, suggesting strong evolutionary constraints for conserving their structures. Although both S- and L-rDNAs share the common overall architecture of IGSs, the most noticeable differences are the relative lengths of the R103b and R103d repeat arrays (Figure 1b).
Are or were?? It change the meaning. When you use "are" it sound like a general information like discussion when you use "were" it gives emphases that they are your results. It is not clear for me what is what.
In general Results part is too long and many sentences are not clear if are your results or you are doing discussion there which is not accurate way of manuscript presenting.
Another examples let's say from M&M
You give figure 10S at the end of the manuscript while those kind of Figures that are obtained at the beginning of work should be presented somehow with results in the beginning.
Line 720 to 721:
To induce DNA hypomethylation, por2 seeds were treated with 200 µM of 5-azacytidine (AzaC) or 50 µM of 9-(2,3-dihydroxypropyl) adenine (DHPA) in water.
For how many days? How many replicates you had? what was the growth conditions? At what point you did sampling? And many other unclear conditions related to the experiment
Please ask someone with more experience to help with manuscript preparation.
Best wishes
Comments on the Quality of English Language
The English language modified grammatically but writing style is messy and not clear. Still many sentences makes no sense.
Author Response
Dear Authors
I appreciate your effort and idea. Your work have good potential but unfortunately your writing style has still big problems. Please note that the English editing services only check the grammar and they can not improve your text content, writing and presenting style. I give you some exmple:
Line 37: which, may, however. How can you use all 3 together???
Response1. Thank you for the adequate reminder. The appropriate change has been made.
You gave the Figure 1 in line 51. However, until line 105 you don’t talk about Tragopogon porrifolius at all. Re organize your introduction. Figure 1b is coming from results and what a figure from results is doing in introduction part?
Response 2. Figure 1a showed the general structure of plant rDNA and, therefore, its reference has been inserted in the Introduction where the rDNA structure is described. However, the removal of the link from the Introduction does not harm the clarity of the article in any way. Thank you.
What does this sentence mean from line 128???
Long PacBio sequences showed that the IGSs in both por1 and popr2 lineages are built of common sequence motifs mainly represented by multiple kinds of repetitive DNA sequences, both tandemly (R) and dispersedly (P-promoters) arranged (Figure 1b).
Response 3. Thank you for pointing out the ambiguity of the sentence. The sentence has been modified.
„We used long PacBio sequences to analyze the structure variability of rDNA units in por1 and por2 lineages of T. porrifolius. The IGSs in both lineages were built of common sequence motifs mainly represented by multiple kinds of tandemly arranged repeats (R) and dispersed promoter elements (Ps) (Figure 1b).“
Another example is from Line 153 to 156
Most of the core promoters (P1, P2, P3, P6, and P7) are identical in both L- and S-rDNAs, suggesting strong evolutionary constraints for conserving their structures. Although both S- and L-rDNAs share the common overall architecture of IGSs, the most noticeable differences are the relative lengths of the R103b and R103d repeat arrays (Figure 1b).
Are or were?? It change the meaning. When you use "are" it sound like a general information like discussion when you use "were" it gives emphases that they are your results. It is not clear for me what is what.
Response 4. Many thanks for this notice. We have attempted to make relevant corrections throughout the text.
In general Results part is too long and many sentences are not clear if are your results or you are doing discussion there which is not accurate way of manuscript presenting.
Response 5. You are right. The results have been shortened and any excess speculation has been removed. Thanks for the relevant comment.
Another examples let's say from M&M
You give figure 10S at the end of the manuscript while those kind of Figures that are obtained at the beginning of work should be presented somehow with results in the beginning.
Response 5. Thanks for the relevant comment. The original Figure S10 has been transferred as Figure S1 to the last paragraph of the Introduction.
Line 720 to 721:
To induce DNA hypomethylation, por2 seeds were treated with 200 µM of 5-azacytidine (AzaC) or 50 µM of 9-(2,3-dihydroxypropyl) adenine (DHPA) in water.
For how many days? How many replicates you had? what was the growth conditions? At what point you did sampling? And many other unclear conditions related to the experiment
Response 6. Thanks for pointing out the lack of information in some methods. The conditions of cultivation of seedlings under the action of epigenetic agents, the time of collection of biological material, the number of biological replicates were introduced.
“Young leaves (whole centrally situated the youngest leaf weighing 80-120 mg), root tips (fresh 5 – 10 mm), and entire seedlings (five days old) were used for comparative transcription and methylation analyses. Leaves and root tips were harvested from five- to six-month-old plants (August, September); one por1 (por-34) and two por2 (por2-15 and por2-134) plants were used as material sources”.
“During germination, the seeds were shaken on a horizontal shaker at 70 rpm, room tem-perature and lighting conditions of 12 hours light and 12 hours dark. The solutions were replaced daily. Total DNAs and RNAs were isolated from each four- to five-day-old seed-ling (three biological replicates)”.
Please ask someone with more experience to help with manuscript preparation.
Response 7. An experienced colleague evaluated the manuscript and his critical comments, mainly regarding the results, were considered.
We believe that all comments have helped to significantly improve the quality of the manuscript.
Reviewer 2 Report
Comments and Suggestions for Authors
The authors have responded to my concerns and I am now happy with the revision of the main part of the MS. But I am still concerned with the abstract. Despite agreeing in the response that the association of L and S rDNA with chromosomes A and D is circumstantial (see also the heading 2.3, line 283), the abstract presents this association as a fact rather than the indirect assignment (as described on line 333). This is misleading.
I also stumbled about the use of ‘elimination’ in respect to por1, in line 19. How do the authors know that por2 is older than por1. I think it should they ‘absence of L-rDNA from D-NORs’ (which is what you describe it as in line 293 of the MS). In the introduction on line 113 it says ‘deleted’ for por1, again indicating that por1 is younger or manipulated. In the discussion some more elaboration is given (line 510ff) and it says ‘Assuming that both negligible D-NORs in por1 arose as a consequence of the homologous deletion of a large portion of the L-rDNAs,…’.
Please correct.
Author Response
The authors have responded to my concerns and I am now happy with the revision of the main part of the MS. But I am still concerned with the abstract. Despite agreeing in the response that the association of L and S rDNA with chromosomes A and D is circumstantial (see also the heading 2.3, line 283), the abstract presents this association as a fact rather than the indirect assignment (as described on line 333). This is misleading.
Response 1. Thank you for the very relevant comments regarding the relationship between both rDNA variants and NORs. Direct statements regarding association of S-rDNA with A-NORs and L-rDNA with D-NORs have been removed from the abstract. Even heading 2.3 we have toned down somewhat.
I also stumbled about the use of ‘elimination’ in respect to por1, in line 19. How do the authors know that por2 is older than por1. I think it should they ‘absence of L-rDNA from D-NORs’ (which is what you describe it as in line 293 of the MS). In the introduction on line 113 it says ‘deleted’ for por1, again indicating that por1 is younger or manipulated. In the discussion some more elaboration is given (line 510ff) and it says ‘Assuming that both negligible D-NORs in por1 arose as a consequence of the homologous deletion of a large portion of the L-rDNAs,…’.
Response 2. You are right that we do not know if there was rDNA deletion in por1 or an amplification of rDNA in por2. Therefore, we have avoided claims about deletion throughout the abstract, introduction and results. The possibility of deletion or amplification are then discussed carefully only in the section Discussion.
Thank you for all the comments which, I believe, significantly improved the quality of the article
Round 3
Reviewer 1 Report
Comments and Suggestions for Authors
Dear Authors
The revised version is more clear now and readable. Still some minor error needs to be fixed.
Title is a little bit confusing: correlates differentially with cytosine methylation OR correlated with differentially cytosine methylation???
Abstract:
Line 13: the latter with more than double the rDNA units distributed between approximately equally sized NORs chromosomes A and D. What does this sentence mean??
Introduction: Where did you refer to figure 1a in the text if you deleted it in line 56?
Line 104 there are two dots (.)
Part of the Figure 3 is not visible
Line 301: what are those drugs? Do they have a name?
Line 316: The two? delete "the"
Line 343: Those methylome marks were in the promoter region or gene bodies?
Discussion: It would be useful to explain why only Por2 seeds were exposed to epigenetic modulation treatment.
Comments on the Quality of English Language
English is ok some very small errors.
Author Response
Comments and Suggestions for Authors
Dear Authors
The revised version is more clear now and readable. Still some minor error needs to be fixed.
Title is a little bit confusing: correlates differentially with cytosine methylation OR correlated with differentially cytosine methylation???
Response 1. You're right that the title was a bit ambiguous, so we've edited it slightly. We believe that it now describes the essence of the article clearly enough. Thank you.
„Transcriptional silencing of 35S rDNA in Tragopogon porrifolius correlates with cytosine methylation in sequence-specific manner“
Abstract:
Line 13: the latter with more than double the rDNA units distributed between approximately equally sized NORs chromosomes A and D. What does this sentence mean??
Response 2: This part of the sentence has been rewritten so we believe it is now much clearer. …..“the latter with more than twice the rDNA copy numbers, distributed approximately equally between NORs on chromosomes A and D“.
Thank you for the important notice.
Introduction: Where did you refer to figure 1a in the text if you deleted it in line 56?
Response 3. Original panels a and b in Figure 1, both showing the rDNA structure have been joined under the a)
In Figure 1, the original panels a) and b), both showing the rDNA structure, are now both labeled as a). The reference to Figure 1a in the introduction has been removed.
Line 104 there are two dots (.)
Response 4. Corrected. Thank you.
Part of the Figure 3 is not visible
Response 5. I apologize for any potential inconvenience with this Figure. However, it seems complete to me. Can you please specify which part is invisible?
Line 301: what are those drugs? Do they have a name?
Response 6. Thank you for the stimulating comment. Ihe individual agents (drugs) were listed in the opening sentence and characterized in more detail in the following ones.
„To assess the role of individual epigenetic modifications in silencing of S-rDNA transcription, por2 seedlings were treated with 5-azacytidine (AzaC), 9-(2,3-dihydroxypropyl) adenine (DHPA), and sodium butyrate (NaBT) known to selectively interfere with respective biochemical pathways“.
Line 316: The two? delete "the"
Response 7. Corrected. Thank you.
Line 343: Those methylome marks were in the promoter region or gene bodies?
Response 8. As mentioned in the text, the methylation analysis concerned the 5‘-ETS2 region, which represents the 5‘-end of the primary rDNA transkript (Figure 1a). In the text, the ETS2 region was somewhat more characterized with respect to the start of transcription
„We focused on the 5’-ETS2 regions representing the 5’- end of the primary transcript (Figure 1a) and containing homogenized specific SNPs allowing discrimination between individual rDNA variants (Figures 4a and S5a) “.
Discussion: It would be useful to explain why only Por2 seeds were exposed to epigenetic modulation treatment.
Response 9. In por2 (persistently active L-rDNA and silent S-rDNA), drug-activated S-rDNAs can be easily distinguished from persistently active L-rDNA by CAPS (Figure 2a, b). Since por1 lineage contains only S-rDNAs, drug-activated S-rDNAs can be discriminated from persistently active S-rDNAs only by sequencing (Figure 2d). Therefore, we have added the following sentence to the end of the paragraph dealing with the action of epigenetic agents: „Because it was difficult to distinguish between drug-activated and persistently active S-rDNAs, the por1 lineage was not subjected to analogous treatments “.
Comments on the Quality of English Language
English is ok some very small errors.